# Analyses of Conditional Knockout Mice for *Pogz*, a Gene Responsible for Neurodevelopmental Disorders in Excitatory and Inhibitory Neurons in the Brain

**DOI:** 10.3390/cells13060540

**Published:** 2024-03-19

**Authors:** Nanako Hamada, Takuma Nishijo, Ikuko Iwamoto, Sagiv Shifman, Koh-ichi Nagata

**Affiliations:** 1Department of Molecular Neurobiology, Institute for Developmental Research, Aichi Developmental Disability Center, 713-8 Kamiya, Kasugai 480-0392, Japan; nhamada@inst-hsc.jp (N.H.); tnishijo@inst-hsc.jp (T.N.); iwamoto@inst-hsc.jp (I.I.); 2Department of Genetics, The Alexander Silberman Institute of Life Sciences, The Hebrew University of Jerusalem, Jerusalem 91904, Israel; sagiv.shifman@mail.huji.ac.il; 3Department of Neurochemistry, Nagoya University Graduate School of Medicine, 65 Tsurumai-cho, Showa-ku, Nagoya 466-8550, Japan

**Keywords:** POGZ, corticogenesis, synapse, conditional knockout (cKO) mice

## Abstract

POGZ (Pogo transposable element derived with ZNF domain) is known to function as a regulator of gene expression. While variations in the *POGZ* gene have been associated with intellectual disabilities and developmental delays in humans, the exact pathophysiological mechanisms remain unclear. To shed light on this, we created two lines of conditional knockout mice for *Pogz*, one specific to excitatory neurons (Emx1-Pogz mice) and the other to inhibitory neurons (Gad2-Pogz mice) in the brain. Emx1-Pogz mice showed a decrease in body weight, similar to total *Pogz* knockout mice. Although the two lines did not display significant morphological abnormalities in the telencephalon, impaired POGZ function affected the electrophysiological properties of both excitatory and inhibitory neurons differently. These findings suggest that these mouse lines could be useful tools for clarifying the precise pathophysiological mechanisms of neurodevelopmental disorders associated with *POGZ* gene abnormalities.

## 1. Introduction

POGZ (Pogo transposable element derived with ZNF domain) is a heterochromatin protein 1 α (HP1α)-binding protein containing a cluster of multiple C2H2-type zinc fingers, a centromere protein (CENP) B-like DNA-binding domain, and a DDE domain that might regulate gene expression [1]. POGZ functions as a transcriptional regulator and influences gene expression by interacting with proteins and DNAs via zinc fingers [1,2]. Since POGZ seems to be involved in chromatin regulation, a process that involves modifying the structure of DNA and histones, this molecule is likely to regulate gene expression by modulating chromatin structure. On the other hand, given the primary expression at the embryonic stages of the mouse brain [3], POGZ is assumed to play a crucial role in neuronal development.

With trio-based whole exome sequencing (WES) and whole genome sequencing, *POGZ* has been identified as a causative gene for White–Sutton syndrome (WHSUS), an autosomal dominant neurodevelopmental disorder (NDD). The clinical spectrum of this syndrome is relatively wide, with known multisystem manifestations, including autism spectrum disorder (ASD), developmental delay, intellectual disability (ID), feeding and gastrointestinal difficulties, seizures, sleep problems, hearing loss, vision problems, and genitourinary abnormalities [4,5,6,7]. In addition, congenital heart disease is reported to be associated with *POGZ* haploinsufficiency in some cases [6,8]. Among a total of 141 cases of WHSUS caused by *POGZ* gene abnormalities [6], 80% were null variants, suggesting that loss of function is the main mechanism of pathogenicity. However, not only the underlying mechanism(s) of *POGZ* gene abnormalities but also the physiological significance of POGZ during brain development remains to be elucidated at the molecular and cell biological levels.

Aiming to elucidate the etiology of POGZ disease, several lines of mouse models were generated. A model with a heterozygous or homozygous nervous system-specific deletion of the *Pogz* gene mimicked several of the human symptoms, such as microcephaly, growth impairment, increased sociability, and learning and motor deficits [9]. Mice with the heterozygous Q1038R, a dominant negative de novo variant of *POGZ* identified in a patient with ASD, exhibited decreased body and brain size and ASD-related behavioral abnormalities [10]. Significantly, complete knockout (KO) of *Pogz* [11] or homozygosity for the Q1038R variation in mice [10] both resulted in early embryonic lethality. Computed tomography (CT) scanning of Q1038R homozygous mouse embryos (E15.5) showed a ventricular septal defect, which was suspected of causing embryonic lethality, suggesting the relationship between congenital heart disease and *POGZ* mutation [6].

In the present study, we generated two lines of conditional KO (cKO) mice for *Pogz* in (1) excitatory neurons in the cerebral cortex and (2) inhibitory neurons in the central nervous system (CNS). We then performed morphological, cell biological, and electrophysiological analyses using these two mouse lines to elucidate the physiological significance of POGZ in excitatory and inhibitory neurons. While the brains from the two lines appeared to be morphologically normal, electrophysiological analyses revealed that impaired POGZ function may be associated with altered synaptic functions, which may be related to the underlying mechanism of neuronal aspects of WHSUS.

## 2. Materials and Methods

### 2.1. Ethics Statement

We followed the fundamental guidelines for the proper conduct of animal experiments and related activity in academic research institutions under the jurisdiction of the Ministry of Education, Culture, Sports, Science, and Technology (Tokyo, Japan). All protocols for animal handling and treatment were reviewed and approved by the Animal Care and Use Committee of the Institute for Developmental Research, Aichi Developmental Disability Center (approval number: 2019-013).

### 2.2. Generation of the POGZ Mutant Model Mice

C57BL/6J mice carrying a floxed (exon 7) *Pogz* allele (*Pogz* fl/fl) [9] were crossed with the *Emx1*-Cre line for dorsal telencephalon-specific KO mice [12] and *Gad2*-Cre [12] line to generate cKO mice specific for CNS-inhibitory neurons. Genotypes were determined by PCR as described [9]. All animals were housed at a temperature of 22–24 °C with 40–60% humidity, under a 12 h light/dark cycle (light on at 07:00, off at 19:00), with free access to food and water. Embryonic day 0.5 (E0.5) was typically defined as noon on the day when a vaginal plug was observed.

### 2.3. Antibodies

Polyclonal rabbit anti-POGZ and anti-Parvalbumin (PV) were generated as described previously [3,13]. Rabbit polyclonal anti-GFP (Medical & Biological Laboratories, Tokyo, Japan, Cat# 598, 1:1000), anti-Cux1 (Gene Tex, Irvine, CA, USA, Cat# GTX56275, 1:300), anti-Pax6 (BioLegend, San Diego, CA, USA, Cat# PRB-278P, 1:500), and anti-phospho-Histone H3 (Ser10) (PHH3) (Cell Signaling, Danvers, MA, USA, Cat# 9701, 1:400) were used. Rat monoclonal anti-Ctip2 was from Abcam (Cambridge, UK, Cat# ab18465, 1:500). Alexa Fluor 488- and 568-labeled IgG were used as secondary antibodies (Abcam, Cat# ab150077, Cat# ab175471, 1:1000). 4′,6-diamidino-2-phenylindole (DAPI) (Sigma-Aldrich, St. Louis, MO, USA, Cat# D9542, 0.2 μg/mL) was used to stain DNA.

### 2.4. Immunohistochemistry

For sampling the brains, mice were deeply anesthetized by inhalation of isoflurane and then perfused with phosphate-buffered saline (PBS) followed by 4% paraformaldehyde (PFA) in PBS. After perfusion, the embryonic and adult brains were dissected out and soaked in 4% PFA for at least 16 h and then sectioned coronally at 100 μm or 12 μm thickness. For immunostaining of embryonic brains, sections were placed onto MAS-coat slides (Matsunami Glass, Osaka, Japan) and treated with HistoVT One (Nacalai Tesque Inc., Kyoto, Japan, Cat# 06380-05) at 70 °C for 20 min. For adult brains, floating sections were incubated in Epitope Retrieval Solution pH 9 (Leica Biosystems, Cat# RE7119-CE) at 50 °C for 3 h. After washing with PBS containing 0.05% Tween (PBST), the sections were blocked with 1% BSA in PBST, and a primary antibody reaction was performed in PBST at 4 °C overnight. Secondary antibody reaction together with nuclear staining with DAPI (0.2 μg/mL) was carried out in PBST for 1 h. Stained sections were mounted with the anti-fading mounting medium (PERMAFLUOR, Cat#TA-030-FM, Thermo Scientific, Waltham, MA, USA). Fluorescent images were captured with an LSM880 confocal laser microscope (Carl Zeiss, Oberkochen, Germany). Bright-field images were captured with a BZ-9000 microscope (Keyence, Osaka, Japan).

### 2.5. Golgi–Cox Staining and Spine Analysis

Golgi–Cox staining was performed using the FD Rapid GolgiStain kit (FD NeuroTechnologies, Columbia, MD, USA) following the manufacturer’s instructions with slight modifications. After mice (one month old, male) were deeply anesthetized with isoflurane and decapitated, brains were quickly harvested and immersed in FD Solution AB (A:B = 1:1) for 2 weeks at room temperature in the dark. After transferring to FD Solution C or tissue-protectant solution (20% Sucrose, 15% glycerol in DDW) and keeping in the dark at 4 °C for 72 h, the brains were sectioned coronally at 100 μm thickness in the tissue-protectant solution. The sections were mounted and stained as described in the instructions. After being dehydrated, the slides were mounted with Permount (Fisher Scientific, Pittsburgh, PA, USA). Z-stack images (20–30) with 1 μm intervals for Golgi-stained dendrites and 0.1-μm intervals for spines were taken using a 20× and 100× lens, respectively, on a BZ-9000 microscope. The branch number and length of basal dendrite were analyzed using NeuronJ, a plugin of Fiji. The number of spines on each apical dendrite within 50–100 μm away from the cell soma was counted using Dendritic Spine Counter, a plugin of Fiji.

### 2.6. In Utero Electroporation

In utero electroporation was performed as previously described [14]. Briefly, after mice were deeply anesthetized with a mixture of butorphanol (5 mg/kg), medetomidine (0.75 mg/kg), and midazolam (4 mg/kg) [15], indicated plasmid was injected into the lateral ventricles of embryos, followed by electroporation using a NEPA21 electroporator (NEPA Gene, Chiba, Japan) with 50 ms of 35 V electronic pulse for 5 times with 450 ms intervals. In this method, each plasmid was introduced into the somatosensory area, which is included in the parietal lobe. Brains were fixed at indicated embryonic or postnatal day, sectioned, and analyzed. All experimental procedures were carried out during the daytime. Animals were neither excluded nor harmed during experiments.

### 2.7. Electrophysiological Analyses

The analyses were performed as described previously [16]. Briefly, coronal cortical slices (300 μm thickness) from P12–19 mice were prepared in ice-cold cutting Krebs solution using a microslicer (PRO7, Dosaka, Kyoto, Japan). The slices were transferred to a holding chamber containing standard Krebs solution and incubated at room temperature. For recording, a slice was superfused with standard Krebs solution at a rate of 3–4 mL/min. To record synaptic currents, patch pipettes made from borosilicate glass capillaries were filled with a CsCl-based internal solution. For current-clamp recordings, patch pipettes were filled with a K-gluconate-based internal solution. Whole-cell recordings were conducted on pyramidal neurons in the cortical layer II/III and inhibitory neurons in the cerebral cortex using a patch-clamp amplifier (Axopatch 200B, Molecular Devices, Foster City, CA, USA) and pCLAMP8 software (ver. 8.2.0.235) (Molecular Devices). Pyramidal neurons were identified both visually and electrophysiologically, while inhibitory neurons were labeled based on the fluorescence signal of EGFP expressed via in utero electroporation at embryonic day 13. Miniature excitatory postsynaptic currents were recorded in the presence of bicuculline (10 μM, Abcam, Cat# ab120110), strychnine (0.5 μM, Tokyo Chemical Industry, Tokyo, Japan, Cat# S0257), and tetrodotoxin (TTX) (0.5 μM, Wako, Cat# 206-11071). Miniature inhibitory postsynaptic currents were recorded in the presence of 6,7-Dinitroquinoxaline-2,3-dione (DNQX) (5 μM, Cayman Chemical Company, Tokyo, Japan, Cat# 14583), D-AP5 (25 μM, Cayman Chemical Company, Cat# 14539), strychnine (0.5 μM) and TTX (0.5 μM).

### 2.8. Statistical Analyses

Cell counting and traces for all cell imaging experiments were assessed in a blinded manner by a technical staff member who was not aware of the experimental conditions. Statistical significance was determined by a non-parametric Mann–Whitney U test using Prism software (GraphPad Prism 7, GraphPad Software Inc., La Jolla, CA, USA). The level of statistical significance was set at *p* < 0.05. Box and whisker plots represent median value (horizontal bars), 25 to 75 percentiles (box edges), and whiskers extend to the largest and smallest observed values that are not outliers, while the cross in the boxes shows the average value.

## 3. Results

### 3.1. Generation of Pogz Conditional KO Mice for Excitatory and Inhibitory Neurons

We generated Pogz cKO mice for excitatory neurons in the cerebral cortex and inhibitory neurons in the CNS by crossing floxed-Pogz mice with Emx1-Cre and Gad2-Cre driver mice, respectively. These mice were termed Emx1-Pogz mice (Emx1^(+/−)^/Pogz^(−/−)^) and Gad2-Pogz mice (Gad2^(+/−)^/Pogz^(−/−)^). Control littermates without Pogz deletion were also prepared.

The gross morphology of the telencephalon of Emx1-Pogz mice and Gad2-Pogz mice was similar to that of control littermates at the young adult stage (Figure 1A). The most obvious phenotype previously described for Pogz-cKO mice, in which homozygous mutations were restricted to the nervous system, was a significant reduction in the body weight relative to control littermates across development [9]. As to the cKO mice we generated, Emx1-Pogz mice demonstrated reduced body weight, whereas Gad2-Pogz mice showed normal body size (Figure 1B,C), suggesting a crucial role of excitatory neurons in the mouse body size.

The efficient deletion of Pogz in excitatory and inhibitory neurons was confirmed in Emx1- and Gad2-Pogz mice, respectively, by immunostaining. POGZ was barely detected in cortical neurons of Emx1-Pogz mice at P0 when compared to those of wild-type mice [3], whereas the protein was strongly detected in medium spiny neurons in the striatum at P30 (Figure 1D, left). As to Gad2-Pogz mice, POGZ was strongly visualized in pyramidal neurons in the upper cortical layers, whereas immunoreactivity was hardly detected in the striatum under the same conditions (Figure 1D, right). In these experiments, immunostaining was performed at the same time. Therefore, POGZ staining in the cortex of Gad2-Pogz mice and in the striatum of Emx1-Pogz mice can be positive controls for the cortex of Emx1-Pogz mice and the striatum of Gad2-Pogz mice, respectively.

### 3.2. Cortical Architecture of Emx1-Pogz Mice in Embryonic and Adult Stages

After birth, dendritic arborization, the cerebral cortex undergoes a series of developmental processes, including dendritic arborization, axon extension, synapse formation, and myelination, to establish mature layer structures in a region-specific manner. We conducted morphological analyses of Emx1-Pogz mice and immunostained their brains with Cux1 (a marker for neurons in layers II to IV) and Ctip2 (a marker for neurons in layer V/VI) at P0 and P30. Since the cell size of Ctip2-positive layer V neurons was larger than that of layer VI neurons, layers II~IV, V, VI, and white matter were roughly assignable in the somatosensory area. Based on the staining patterns of Ctip2 and Cux1, the layer structure of the cerebral cortex of this line did not differ significantly from that of control littermates at these time points (Figure 2A).

During brain development, cortical excitatory neurons migrate from the ventricular zone (VZ) toward the pial surface. This process is frequently affected by blocking the expression of genes involved in brain development [17]. Although the cortical layer structure appeared to be normal in Emx1-Pogz mice, neuronal migration might be affected during corticogenesis. To test this possibility, Emx1-Pogz mice were crossed, and the resulting embryos were electroporated in utero with pCAG-EGFP (0.5 μg) at E14 and fixed at P0. When the distribution of EGFP-positive cells was examined, most of them migrated to the superficial layer of the cortical plate (CP), just like the control mice (Figure 2B). Hence, the radial migration of cortical excitatory neurons is thought to be normal in Emx1-Pogz mice.

We then analyzed interhemispheric axon projections of cortical neurons in Emx1-Pogz mice in vivo. pCAG-EGFP (0.5 μg) was electroporated in utero into control embryos at E14, fixed at P16, and the sections were stained for GFP. The results showed that axon bundles extended normally into the contralateral hemisphere (Figure 2C, left panels), indicating the successful development of interhemispheric connections. Similarly, axons from POGZ-deficient cortical neurons extended appropriately into the cortical layers on the contralateral side after reaching the contralateral white matter at P16 (Figure 2C, right panels). These observations strongly suggest that there is no significant defect in the cortical development in Emx1-Pogz mice.

### 3.3. Cortical Architecture of Gad2-Pogz Mice

The cortical layer structure of Gad2-Pogz mice did not show any significant differences from that of control littermates when immunostained for Cux1 and Ctip2 at P30 (Figure 3A). This finding is consistent with the observations made in Emx1-Pogz mice (Figure 2A). To investigate whether the deficiency of *Pogz* in inhibitory neurons affects their generation and distribution in the cerebral cortex, we examined the cortical architecture of this line. Staining for phospho-histone H3 (pHH3) in E13 cortical slices revealed that the distribution pattern and number of pHH3-positive mitotic cells in the medial ganglionic eminence (MGE) were comparable to those of control mice (Figure 3B). Furthermore, we observed that the distribution and number of PV-positive inhibitory neurons were minimally affected in the cerebral cortex (Figure 3C) and hippocampus (Figure 3D) at P30. Our investigation of the density and distribution of medium spiny neurons, which are inhibitory neurons dominantly distributed in the striatum, revealed that they were normal and similar to the control littermate (Figure 3E).

### 3.4. Role of POGZ in the Dendritic Arbor Development In Vivo

Given that the principal neurological manifestations associated with *Pogz* gene abnormalities, such as ID and ASD, are likely linked to deficiencies in synapse network formation and/or maintenance, an analysis was conducted on the morphology of dendrites and synapses in Golgi-stained pyramidal neurons located in layer II/III of the somatosensory cortices of Emx1-Pogz mice at P30. The study found that there were no significant differences in the length and branch numbers of basal dendrites between Emx1-Pogz mice and their control littermates (Figure 4A–C), indicating that the genetic modification did not have a significant impact on these parameters. Additionally, the study quantified the dendritic spines of layer II/III pyramidal neurons in Emx1-Pogz mice and found no significant alterations were observed in spine density within the somatosensory area when compared to the control mice (Figure 4D,E). These results suggest that the genetic modification did not have a significant effect on dendritic spine density in this region. Overall, the study provides valuable insights into the impact of the *Emx1*-*Pogz* genetic modification on neuronal morphology and function.

We then conducted the same analyses as described above, focusing on Gad2-Pogz mice. Our findings indicate that the absence of POGZ in inhibitory neurons within the CNS had minimal effects on the total length and branch number of basal dendrites in pyramidal neurons located in cortical layer II/III of the somatosensory area when compared to control littermates (Figure 4F–H). Furthermore, it was observed that there was no significant impact on the spine densities of layer II/III pyramidal neurons in the model mice (Figure 4I,J). These findings suggest that the deficiency of *Pogz* in inhibitory neurons has limited influence on the morphology of excitatory neurons in the cerebral cortex. In general, it appears that functional defects in POGZ in either excitatory or inhibitory neurons are unlikely to significantly affect dendritic and synaptic morphologies.

### 3.5. Role of POGZ in Synaptic Transmission in the Cortical Neurons

While ID and ASD are often referred to as synaptic disorders, our analyses have revealed that there are only minimal morphological differences in the development of dendrites and spines in cortical neurons (Figure 4). Our focus therefore shifted to investigating the effects of Pogz deficiency on synaptic functions and neuronal activity, with an emphasis of neuron-type-specific effects. To achieve this, we performed electrophysiological analyses using the whole-cell patch clamp technique with P12–19 mice.

When the synaptic transmission was analyzed in layer II/III pyramidal neurons of Emx1-Pogz mice using voltage-clamp recording, the frequency of miniature excitatory postsynaptic currents (mEPSCs) was significantly lower than that of the control littermates (Figure 5A,B). However, there was no significant difference in mEPSC amplitude between the control and Emx1-Pogz mice (Figure 5A,C). When recording miniature inhibitory postsynaptic currents (mIPSCs), both frequency and amplitude were similar between the control and Emx1-Pogz mice (Figure 5D–F). Subsequently, the excitability of layer II/III pyramidal neurons was analyzed by recording their firing rates via current-clamp recording. The experiment conducted using a 120 pA current pulse to evoke action potentials for 800 ms showed no significant difference between the two groups (Figure 5G,H).

As a next set of experiments, we analyzed synaptic transmission in cortical neurons of Gad2-Pogz mice. We recorded synaptic currents from inhibitory neurons in the cerebral cortex (Figure 6A–F) and from layer II/III pyramidal neurons (Figure 6G–I). The frequency of mEPSCs in Gad2-Pogz mice was significantly higher than that in the control littermates (Figure 6A,B), while there was no significant difference in mEPSC amplitude between the two groups (Figure 6A,C). The Gad2-Pogz mice exhibited a significantly higher frequency of mIPSCs compared to the control mice (Figure 6D,E), whereas mIPSC amplitude was comparable to the control mice (Figure 6D,F). In layer II/III pyramidal neurons, the frequency of mIPSCs was also significantly higher in the Gad2-Pogz mice than in the control mice (Figure 6G,H). However, there was no significant difference in mIPSC amplitude between the two groups (Figure 6G,I).

## 4. Discussion

Heterozygous mutations in the *POGZ* gene have been associated with WHSUS. These variations generally involve deletions, frameshift variations, or other alterations that disrupt the normal function of the *POGZ* gene. The importance of POGZ in cognitive function has also been implicated by analyses with the *Drosophila* knockdown model of the *POGZ* ortholog, row since neuron-specific knockdown of row caused deficits in habituation, a form of non-associative learning highly relevant to both ID and ASD [5].

When POGZ was knocked out in both excitatory and inhibitory neurons within the neuronal system, the mice exhibited notable phenotypes, such as dwarfism, microcephaly, and motor deficits [9]. In stark contrast, Emx1-Pogz and Gad2-Pogz mice did not manifest morphological anomalies in the telencephalon, indicating that the pathogenicity of WHSUS relies critically on the functional defects of POGZ in both excitatory and inhibitory neurons within the CNS.

While dendritic and synaptic morphology remained normal in Emx1-Pogz and Gad2-Pogz mice, electrophysiological analyses revealed distinct impacts on synaptic functions in these lines. Considering the shared pathophysiological mechanisms encompassing ID to ASD, we hypothesize that structural and/or functional defects in synapses may contribute to the etiology of WHSUS. Conversely, the regulatory role of POGZ in mitosis suggests its involvement as a transcriptional regulator in the control of progenitor proliferation, potentially explaining the frequent occurrence of microcephaly and dysmorphia, including craniofacial and limb development anomalies, in patients with POGZ variations. However, mouse lines specific to either cerebral excitatory neurons or inhibitory neurons had weaker effects on brain and body sizes than the nervous system-specific cKO mice. We assume that concurrent haploinsufficiency effects within the *POGZ* gene in both excitatory and inhibitory neurons are essential for the pathogenicity of POGZ disease.

Given that the dendritic development and spine density of Emx1-Pogz mice appear to be normal (Figure 4), the results of the electrophysiological analyses (Figure 5) suggest that *Pogz* deficiency in cortical excitatory neurons reduces presynaptic glutamate release, leading to a decrease in excitatory transmission in excitatory neurons without affecting inhibitory synaptic transmission. On the other hand, the findings on Gad2-Pogz mice suggest that *Pogz* deficiency in inhibitory neurons may disturb the excitation-inhibition (E–I) equilibrium by elevation of synapse formation and/or presynaptic GABA release in cortical inhibitory neurons. In addition, *Pogz* deficiency in inhibitory neurons could heighten inhibitory transmission in excitatory neurons by an increase in presynaptic GABA release. Taken together, the functional defects in POGZ may cause suppression of pyramidal neurons and disrupt the E–I balance in inhibitory neurons in the cerebral cortex. While the observed phenotypes may reflect the pathophysiology of ASD and other NDDs associated with *POGZ* gene abnormalities, further analyses of the cell morphology and synaptic morphology/number of inhibitory neurons are crucial to better understand the pathophysiological significance of *POGZ* disease.

Electrophysiological analyses were also conducted in POGZ^WT/Q1038R^ mice, a model of a de novo variant identified in an ASD patient [10]. The results showed a significant increase in the frequency of mEPSCs in layer II/III pyramidal neurons, indicating hyperactivation. It is possible that this increase is due to an increase in dendritic spine density. On the other hand, dendric spine density remains unchanged in Emx1-Pogz mice, and thus, the decrease in mEPSC frequency may be a result of a reduction in presynaptic glutamate release. The difference in our findings and those of POGZ^WT/Q1038R^ mice could be attributed to variations in dendric spine phenotypes. While there may be a reduction in presynaptic glutamate release in POGZ^WT/Q1038R^ mice, it is plausible that this effect is concealed by an increase in dendric spine density.

In the cortical inhibitory neurons, Gad2-Pogz mice exhibited a significantly higher frequency of mIPSCs, while the amplitude was similar to that of the control mice (Figure 6E,F). Conversely, in the cerebellar inhibitory neurons (Purkinje cells) of nervous system-specific Pogz KO mice, the frequency of spontaneous IPSCs remained unaltered while the amplitude increased [9]. The observed difference can possibly be attributed to the distinct signaling networks used in the cerebral cortex and cerebellum and/or their susceptibility to *Pogz* defects. *Pogz* deficiency appears to lead to an increase in inhibitory input to cortical inhibitory neurons and Purkinje cells.

While the exact molecular mechanisms behind the pathophysiological significance of *POGZ* gene abnormalities are not yet fully understood, the loss of POGZ function is possible to disrupt the proper regulation of downstream signaling. To explore the connection between chromatin, transcriptional, and synaptic dysregulation and cognitive deficits associated with NDDs, genome-wide transcriptional changes induced by Pogz-knockdown were examined using RNA-seq analyses of prefrontal cortex tissue of juvenile mice [18]. Consequently, Pogz deficiency was found to lead to upregulation of genes enriched in neuroinflammation [18], which is similar to the elevation of pro-inflammatory genes observed in humans with NDDs [19,20,21]. Pogz deficiency was also observed to cause a significant increase in pro-phagocytic microglial activation in the prefrontal cortex [18]. In addition, a significant reduction in glutamatergic transmission and postsynaptic protein expression was detected [18]. These findings suggest that POGZ may have functions related to NDDs, offering a possible explanation that connects chromatin, transcriptional, and synaptic dysregulation to cognitive deficits associated with NDDs. Upon examining the functional roles of the differentially expressed genes (DEGs) using gene ontology (GO) pathway analysis, the major pathways downregulated were those for action potential and membrane depolarization [18]. These findings may provide an explanation for the aberrant electrophysiological properties of neurons in Emx1- and Gad2-Pogz mice. On the other hand, downregulated DEGs due to Pogz deficiency were over-enriched in neuronal genes, which may indicate the decreased transcription of some neuron-related genes [18]. This is consistent with the finding that POGZ may promote the transcription of clustered synaptic genes during early development [22]. It is suggested that synaptic functions that have not yet been analyzed may be affected in Pogz-deficient excitatory and inhibitory neurons. Further analysis is necessary to address this issue.

Although the specific downstream pathways affected by *POGZ* gene abnormalities remain unknown, *AURKB*, encoding Aurora B kinase and an interactive partner for POGZ, emerges as a potential research target [1]. Notably, *AURKB* has been implicated in another autosomal dominant neurological disorder, spinocerebellar ataxia type 10 (SCA10), characterized by cerebellar dysfunction and varying degrees of signs from other components of the CNS [23]. Further studies are necessary to provide insight into the underlying pathophysiological mechanisms of POGZ disorders, including WHSUS, and to identify novel therapeutic targets to interfere with the pathogenesis of the disease.

## Figures and Tables

**Figure 1 cells-13-00540-f001:**
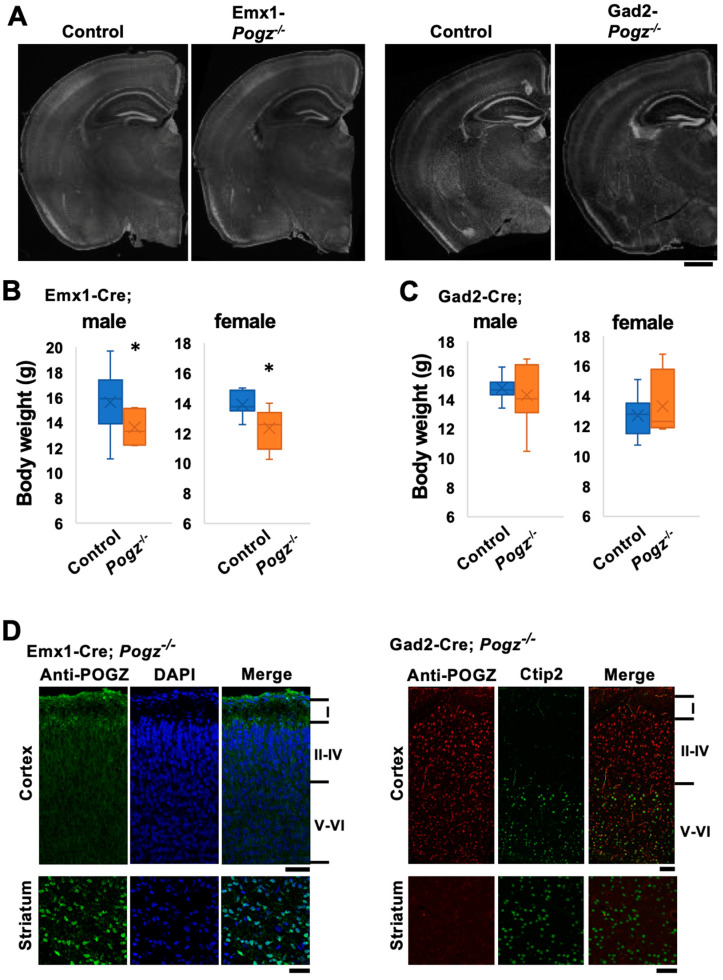
Characterization of the mouse lines with conditional Pogz deficiency in excitatory and inhibitory neurons in the cerebral cortex. (**A**) Coronal sections of the brains of Emx1- and Gad2-Pogz mice and the control littermate at P30 were stained for DAPI. (**B**,**C**) Body weight of Emx1- and Gad2-Pogz mice (male and female) was compared at P30 with that of control littermates. *n* = 14 and 7 for Control and Emx1-Pogz male, respectively, and *n* = 8 and 7 for Control and Emx1-Pogz female, respectively (**B**). *n* = 9 and 8 for Control and Gad2-Pogz male, respectively, and *n* = 9 and 10 for Control and Gad2-Pogz female, respectively (**C**). For detailed information of box plots, see “Statistical analyses” in the Section 2. * *p* < 0.05, Mann–Whitney U test. (**D**) Coronal brain sections of the somatosensory area (P0 or P30 for the cortex of Emx1-Pogz or Gad2-Pogz, respectively) and striatum (P30) of Emx1-Pogz (left panels) and Gad2-Pogz (right panels) mice were stained with anti-POGZ (green) plus DAPI (blue) (left panels) or anti-POGZ (red) plus anti-Ctip2 (green) (right panels). Scale bars, 500 μm (**A**) and 50 μm (**D**).

**Figure 2 cells-13-00540-f002:**
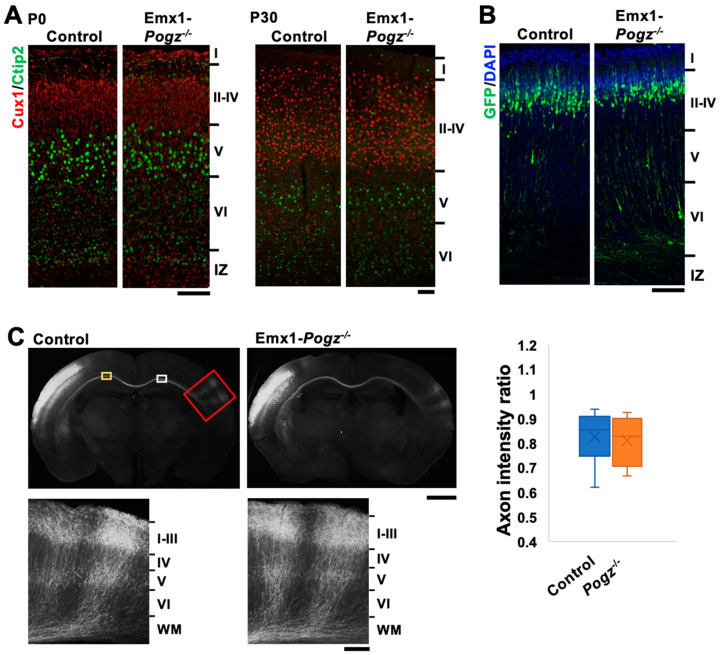
Architecture of cerebral cortex of Emx1-Pogz mice. (**A**) Layer structure of cerebral cortices at P0 (**left**) and P30 (**right**). Coronal sections at the somatosensory area were stained for Ctip2 (green) and Cux1 (red). (**B**) Cortical neuron migration and positioning. pCAG-EGFP (0.5 μg) was electroporated in utero into the VZ progenitor cells at E14. Coronal sections were prepared at P0 and double-stained with anti-GFP (green) and DAPI (blue). (**C**) Axon extension from ipsilateral to contralateral cortex during corticogenesis in vivo. pCAG-EGFP (0.5 μg) was electroporated as in (**A**). Coronal sections were prepared at P16 and stained with anti-GFP (green). Red-boxed areas were magnified to show axon extension into the cortex. The ratio of the GFP intensity in the area (yellow) of the ipsilateral cortex to that in the area (white) of the contralateral cortex was calculated as the axon intensity ratio (n = 6 for each genotype). For detailed information of box plots, see “Statistical analyses” in the Section 2. Scale bars: 100 μm (**A**,**B**), 1 mm (**C**, upper), and 200 μm (**C**, lower).

**Figure 3 cells-13-00540-f003:**
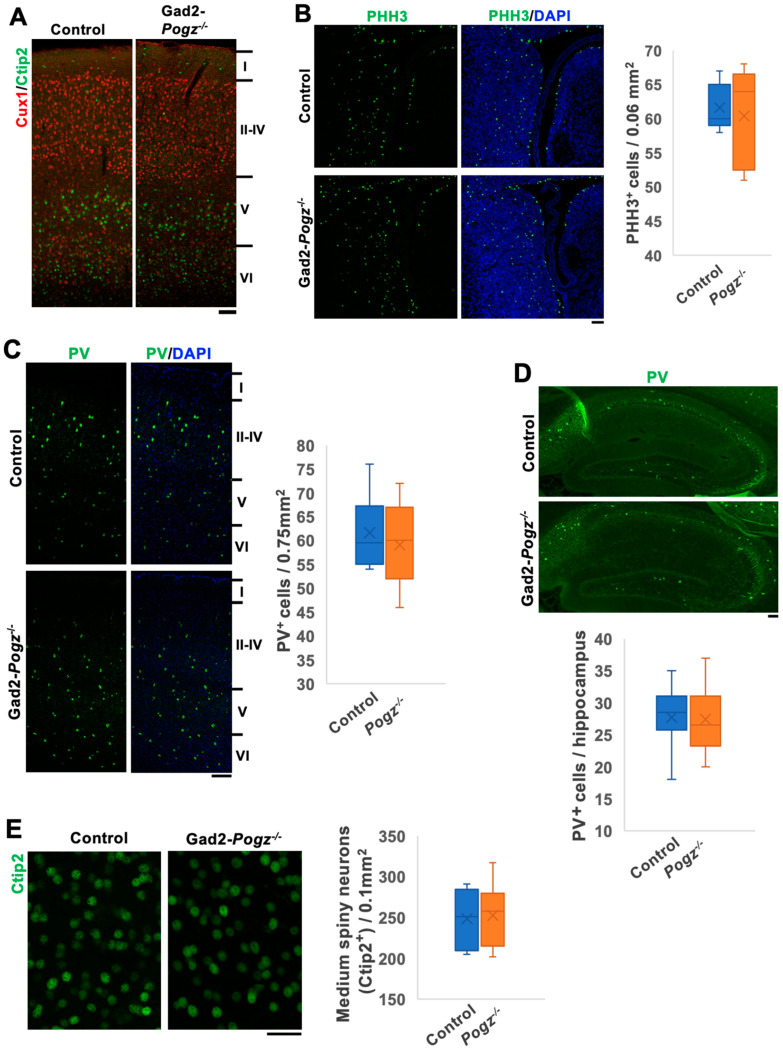
Architecture of telencephalon of Gad2-Pogz mice. (**A**) Layer structure of cerebral cortices at P30. Coronal sections at the somatosensory area from Gad2-Control and Gad2-Pogz mice were stained for Ctip2 (green) and Cux1 (red). (**B**) Sections from Control and Gad2-Pogz mice at E13 were co-stained with anti-pHH3 (green) and DAPI (blue). Quantification analyses: box-and-whisker plots show the number of pHH3^+^ mitotic cells per unit area (0.06 mm^2^) in the MGE from the indicated genotypes. (**C**,**D**) Representative images of cerebral cortex (**C**) and hippocampus (**D**) at P30. Sections from each genotype were co-stained with anti-parvalbumin (PV, green) and DAPI (blue). Quantification analyses: box-and-whisker plots show the number of PV^+^ interneurons per unit area (0.75 mm^2^) (**C**) and the whole area of hippocampus (**D**) from the indicated genotypes. (**E**) Medium spiny neurons in the striatum were stained with Ctip2 (green). Ctip2^+^ neurons were quantified. Box-and-whisker plots show the number of Ctip2^+^ cells per unit area (0.1 mm^2^) from the indicated genotypes. For all quantification, *n* = 5 animals per genotype were used. Two slices were prepared per animal, and 10 fields of views were scored from them. For detailed information of box plots, see “Statistical analyses” in the Section 2. Scale bars; 100 μm (**A**), 50 μm (**B**–**D**), 20 μm (**E**).

**Figure 4 cells-13-00540-f004:**
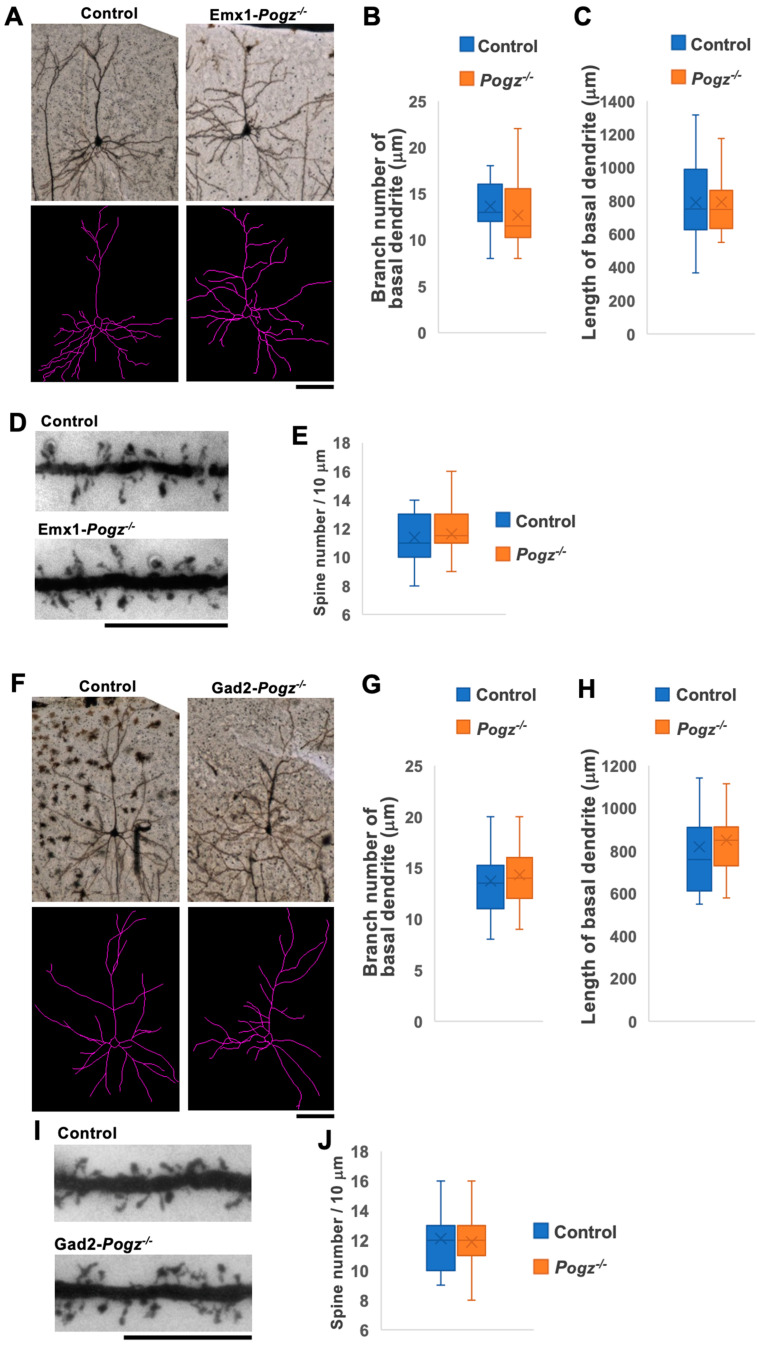
Morphological analyses of dendrites and spines of cortical neurons at P30. (**A**) Representative images of Golgi-stained pyramidal neurons in the somatosensory area of cerebral cortices of Control and Emx1-Pogz mice (**upper** panels). Neuron-tracing images by Image J software (ver. 2.9.0/1.53t) were also shown (**lower** panels). (**B**,**C**) Quantification of (**A**). Branch number (**B**) and length (**C**) of basal dendrites were quantified. (**D**) Representative images of Golgi-stained spines of apical dendrites of layer II/III pyramidal neurons of Control and Emx1-Pogz mice. (**E**) Quantification of spine density in (**D**). *n* = 5 animals per genotype were used. After five slices were prepared per animal, 17 and 20 neurons (**B**,**C**) and 34 and 38 neurons (**E**) were scored from these slices for Control and Emx1-Pogz mice, respectively. (**F**) Representative images for Control and Gad2-Pogz mice as in (**A**). (**G**–**J**) Quantification analyses were conducted as in (**B**–**E**). *n* = 5 animals per genotype were used. After five slices were prepared per animal, 18 and 23 neurons (**G**,**H**) and 34 and 39 neurons (**J**) were scored from these slices for Control and Gad2-Pogz mice, respectively. For detailed information of box plots, see “Statistical analyses” in the Section 2. Scale bars: 50 μm (**A**,**F**), 10 μm (**D**,**I**).

**Figure 5 cells-13-00540-f005:**
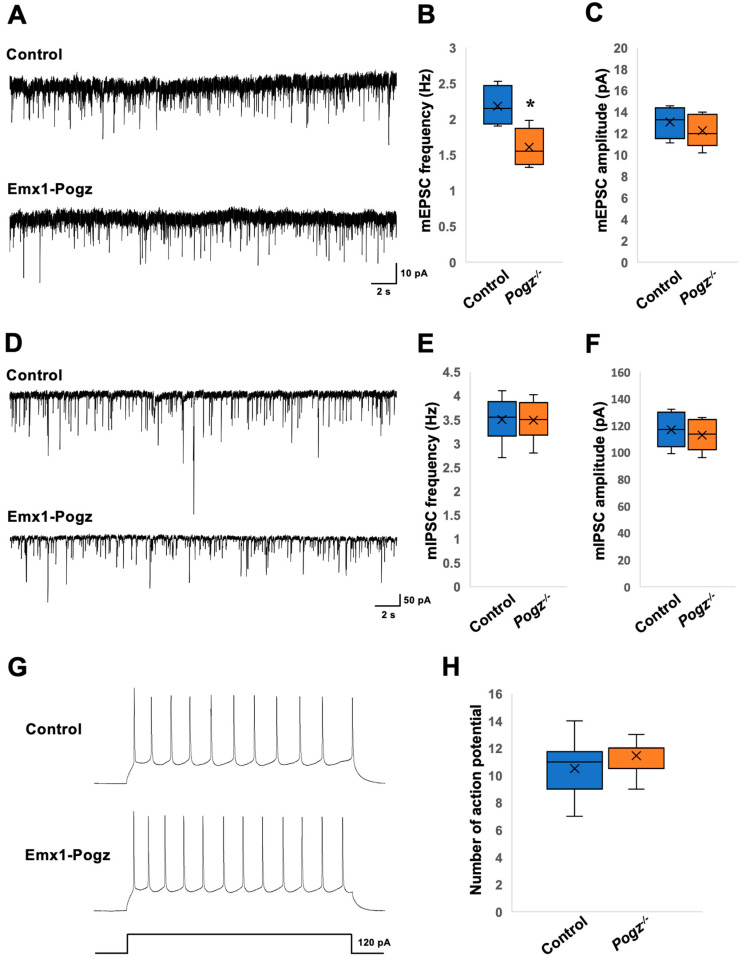
Electrophysiological analyses in layer II/III pyramidal neurons of Emx1-Pogz mice. Postsynaptic currents were recorded from pyramidal neurons in layer II/III at a holding potential of −60 mV. (**A**–**C**) Consecutive traces of mEPSCs were recorded in the presence of bicuculline (a GABA_A_ receptor antagonist, 10 μM), strychnine (a glycine receptor antagonist, 0.5 μM), and tetrodotoxin (TTX) (a sodium channel blocker, 0.5 μM) from Control and Emx1-Pogz mice. For the quantification of mEPSC frequency (**B**) and amplitude (**C**), Control mice were analyzed for 5 neurons in 4 slices from 2 mice, while Emx1-Pogz mice were analyzed for 6 neurons in 4 slices from 2 mice. (**D**–**F**) Consecutive traces of mIPSCs were recorded in the presence of 6,7-Dinitroquinoxaline-2,3-dione (DNQX) (a non-NMDA receptor antagonist, 5 μM), D-AP5 (an NMDA receptor antagonist, 25 μM), strychnine (0.5 μM) and TTX (0.5 μM). For the quantification of mIPSC frequency (**E**) and amplitude (**F**), Both control and Emx1-Pogz mice were analyzed for 6 neurons in 3 slices from 3 mice. (**G**,**H**) Action potentials evoked by a 120 pA current injection for 800 ms (**G**) and quantification of the number (**H**). Control mice were analyzed for 10 neurons in 4 slices from 2 mice, while Emx1-Pogz mice were analyzed for 11 neurons in 6 slices from 3 mice. For detailed information of box plots, see “Statistical analyses” in the Section 2. * *p* < 0.05, Mann–Whitney U test.

**Figure 6 cells-13-00540-f006:**
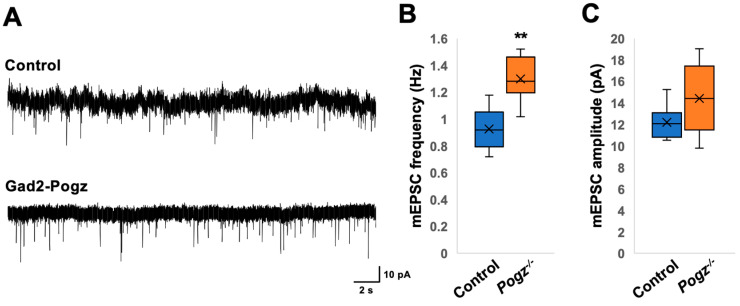
Electrophysiological analyses in neurons of Gad2-Pogz mice. Postsynaptic currents were recorded from cortical inhibitory neurons (**A**–**F**) and pyramidal neurons in layer II/III (**G**–**I**) at a holding potential of −60 mV. (**A**–**C**) Consecutive traces of mEPSCs were recorded in the presence of bicuculline, strychnine, and TTX from Control and Gad2-Pogz mice. For the quantification of mEPSC frequency (**B**) and amplitude (**C**), Both control and Gad2-Pogz mice were analyzed for 6 neurons in 4 slices from 3 mice. (**D**–**F**) Consecutive traces of mIPSCs were recorded in the presence of DNQX, D-AP5, strychnine, and TTX. For the quantification of mIPSC frequency (**E**) and amplitude (**F**), Control mice were analyzed for 6 neurons in 4 slices from 3 mice, while Gad2-Pogz mice were analyzed for 5 neurons in 5 slices from 4 mice. (**G**–**I**) Consecutive traces of mIPSCs were recorded in the presence of DNQX, D-AP5, strychnine, and TTX. For the quantification of mIPSC frequency (**H**) and amplitude (**I**), Control mice were analyzed for 7 neurons in 4 slices from 2 mice, while Gad2-Pogz mice were analyzed for 7 neurons in 5 slices from 3 mice. For detailed information of box plots, see “Statistical analyses” in the Section 2. * *p* < 0.05, ** *p* < 0.01, Mann–Whitney U test.

## Data Availability

Data supporting the findings of this study are available from the corresponding author upon request.

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
