# Peer review of "Analyses of Conditional Knockout Mice for Pogz, a Gene Responsible for Neurodevelopmental Disorders in Excitatory and Inhibitory Neurons in the Brain"

_cells, 2024, doi:10.3390/cells13060540_

Round 1

Reviewer 1 Report

Comments and Suggestions for Authors

Review “Analyses of conditional knockout mice for Pogz, a responsible 2 gene for neurodevelopmental disorders, in excitatory and inhibitory neurons in the brain” by Hamada et al.

The study concerns with the mechanisms of function of POGZ, a heterochromatin binding protein that has been related to the neurologic White-Sutton syndrome. Authors conducted an important and interesting experimental work by using mice that were selectively Knock-Out for POGZ in excitatory and inhibitory neurons.  The study is covered by several scopes, the results are sound and original.

The main findings being that the absence of POGZ is not related to excitatory cell migration from ventricular site to dorsal zones during critical developmental processes. A similar effect was observed when axon elongation was assessed, and inter hemispheric connection was spared in Excitatory neurons-POGZ deficient mice. The genetic deletion of POGZ in excitatory or inhibitory neurons did not show any changes in the architectural spine arborization or genetic control in neuronal expression, therefore, authors conclude that apparently POGZ was not involved in events related to architectural alterations in brain.  Importantly, they found that the electrophysiology of excitatory neurons was affected when in POGZ was deleted from excitatory neurons. They specifically observed a decrease in frequency of miniature excitatory postsynaptic currents (mEPSCs) in layers II/III of cortex, corresponding to pyramidal glutamatergic cells. They also found that in mice lacking POGZ in inhibitory neurons, the frequency of miniature inhibitory postsynaptic currents was increased as compared to neurons from control mice.

The study is well conducted, well written and results are relevant. However, some issues should be corrected, I believe that the most important concerns with statistical analysis.

Major

Under methods, authors state that “Data were not assessed for normality in this study. No test for outliers was conducted in this study. Because data were not assessed for normality, we did not describe and justify any normalization, which did not result in the need for non-parametric analysis.”

A regular procedure to compare groups of scale variables is to assess for data distribution in order to decide whether data in the different groups should be statistically compared with parametric or non-parametric test. (Daniel, 1995). Therefore, it is odd that authors simply decided not to assess normality test and proceed to Welch´s test for comparison; since Welch´s test is based on the assumption that data involved are normally distributed and considers that variance from the groups are different (Welch, 1947). If data did not distributed normally, options are data transformation or non-parametric test.. I believe that authors should look for advice from a statistician to solve this problem.

Minor

Authors worked with mice that were genetically modified to generate POGZ KO mice for excitatory glutamatergic neurons, Emx1-Cre line for dorsal telencephalon-specific KO mice and POGZ KO mice Gad2-Cre line for inhibitory gabaergic neurons. On the resulting progenies they performed morphological and electrophysiological studies on mice after birth.  In some of the experiments (figures 3 and 4), authors report that they performed assays with n=5. It is important that the provide information that this number refers that the brain tissue the different preparations were from different subjects and furthermore that individuals came from different dam, or if they were siblings, it should be stated so.

On page 3 line 136 authors state “mice were deeply anesthetized with the mixture of 3 drugs as mentioned earlier, indicated”. However they did not mention an early procedure of anesthesia induced by 3 drugs. They may correct this issue or specify the drugs and doses used.

On section “2.8 statistical analyses”

Authors begin with the following sentence “No statistical methods were used to predetermine sample sizes, but our sample sizes are similar to those generally employed in the field” I would suggest to delete the text.

Also, in the following sentence, it can be reduced to  “Cell counting and traces for all cell imaging experiments were assessed in  a blinded manner by a technical staff member who was not aware of the experimental conditions”.

In none of the graphs depicted in the manuscript authors use media ± SD. In all cases they used boxplots, therefore the sentence “Results are expressed as means ± SD” is inaccurate and confusing.

On legend to figures 5 and 6, authors refer that they performed student t-test. However, in Methods section they refer that they used Welch´s test for comparison. Although both test are very related, names should be consistent along the manuscript.

On results last paragraph of results section, page 12 lines 386 to 392 the statements written by authors are rather speculative and they correspond to discussion section.

References

 Daniel Wayne W,  Biostatistics: a foundation for analysis in the health sciences. Sixth edition, Published in 1995 in New York by Wiley & Sons., ISBN: 0471854018.

 WELCH BL., The generalisation of student's problems when several different population variances are involved.. Biometrika. 1947;34(1-2):28-35. doi: 10.1093/biomet/34.1-2.28.

Author Response

Reviewer#1 

Major

1) Under methods, authors state that “Data were not assessed for normality in this study. No test for outliers was conducted in this study. Because data were not assessed for normality, we did not describe and justify any normalization, which did not result in the need for non-parametric analysis.”

A regular procedure to compare groups of scale variables is to assess for data distribution in order to decide whether data in the different groups should be statistically compared with parametric or non-parametric test. (Daniel, 1995). Therefore, it is odd that authors simply decided not to assess normality test and proceed to Welch´s test for comparison; since Welch´s test is based on the assumption that data involved are normally distributed and considers that variance from the groups are different (Welch, 1947). If data did not distribute normally, options are data transformation or non-parametric test. I believe that authors should look for advice from a statistician to solve this problem.

> According to the Reviewer’s comment, we have obtained suggestion from a researcher who knows statistics. Based on his suggestion, we conducted a reevaluation using the non-parametric Mann-Whitney U test, which does not require normality assumption, because data were not assessed for normality in this study. According to this change, we rewrote the subsection “2.8 Statistical analyses” in the revised manuscript. Accordingly, we amended legends for Figures 1, 5 and 6.

Minor

1) Authors worked with mice that were genetically modified to generate POGZ KO mice for excitatory glutamatergic neurons, Emx1-Cre line for dorsal telencephalon-specific KO mice and POGZ KO mice Gad2-Cre line for inhibitory gabaergic neurons. On the resulting progenies they performed morphological and electrophysiological studies on mice after birth.  In some of the experiments (figures 3 and 4), authors report that they performed assays with n=5. It is important that the provide information that this number refers that the brain tissue the different preparations were from different subjects and furthermore that individuals came from different dam, or if they were siblings, it should be stated so.

> We agree to the Reviewer’s comment. We rewrote the figure legends and described the numbers of mice, samples and cells analyzed (p. 8, line 291 – 293; p. 10, line 313 – 316 and line 317 – 320 in the revised manuscript).

2) On page 3 line 136 authors state “mice were deeply anesthetized with the mixture of 3 drugs as mentioned earlier, indicated”. However they did not mention an early procedure of anesthesia induced by 3 drugs. They may correct this issue or specify the drugs and doses used.

> We agree to the Reviewer’s comment, and specified the 3 drugs and doses used, and cited a reference #15 (p. 3, line 136 - 137 in the revised manuscript).

3) On section “2.8 statistical analyses”

Authors begin with the following sentence “No statistical methods were used to predetermine sample sizes, but our sample sizes are similar to those generally employed in the field” I would suggest to delete the text.

Also, in the following sentence, it can be reduced to “Cell counting and traces for all cell imaging experiments were assessed in a blinded manner by a technical staff member who was not aware of the experimental conditions”.

> The Reviewer's suggestions would be greatly appreciated. The sentences have been amended to reflect the change in statistical methods used in the '2.8 Statistical Analyses' subsection of the revised manuscript (p. 4, line 166 - 173).

4) In none of the graphs depicted in the manuscript authors use media ± SD. In all cases they used boxplots, therefore the sentence “Results are expressed as means ± SD” is inaccurate and confusing.

> We agree to the Reviewer’s comment. We deleted the sentence “Results are expressed as means ± SD.” in the '2.8 Statistical Analyses' subsection in the revised manuscript.

5) On legend to figures 5 and 6, authors refer that they performed student t-test. However, in Methods section they refer that they used Welch´s test for comparison. Although both tests are very related, names should be consistent along the manuscript.

> We agree to the Reviewer’s comment. We standardized the statistical analysis method to the Mann-Whitney U test throughout the manuscript. The statistical processing of Figure 5 and 6 was also conducted using the Mann-Whitney U test. Consequently, Figure 5B, C, E, F, and H, as well as Figure 6B, C, E, F, H, and I, were replaced in the revised manuscript. Additionally, we revised the description in the subsection '2.8 Statistical Analyses' and legends for Figures 1, 5, and 6 in the new manuscript.

6) On results last paragraph of results section, page 12 lines 386 to 392 the statements written by authors are rather speculative and they correspond to discussion section.

> We agree to the Reviewer’s comment. Since the sentences (p.12 lines 386 - 392 in the old manuscript) are speculative, we deleted them in this section, slightly modified, and moved to the Discussion section in the revised manuscript (p. 14, line 438 - 445).

Reviewer 2 Report

Comments and Suggestions for Authors

The chromatin gene POGZ is among the most commonly de novo mutated gene associated with intellectual disability and autism spectrum disorder. To model this disorder, several groups have generated mutant mouse lines, but the papers describing these mice report somewhat different phenotypes, and much remains to be learned about POGZ deficiency and its link to neurodevelopment.

Here, Hamada et al. generate conditional mutants using Emx1-Cre and Gad2-Cre, in order to ablate POGZ in excitatory vs. inhibitory lineages. The authors' work culminates in some interesting electrophysiological experiments, but the authors additionally did high quality histological characterization of the mutant brains. The physiological experiments suggest that POGZ regulates EPSCs and IPSCs in cortical networks. Interestingly, the authors' analyses suggest that these changes are not mediated by alterations in synapse numbers.

In general, the paper is well written (although the text needs further development in places), and the work is of high quality in general. I have very little to criticize in the authors' manuscript, except that I think it is too short, especially in the Discussion section. I think it should be a good contribution to the field.

Here are my concerns - all minor:

With respect to the study design, I only spotted a few small issues. 

1) "Control" is not well-defined in the text.  Did the authors pool conditional conditional heterozygotes with wild-type?  If so, are they sure that conditional heterozygotes do not have a phenotype?  It strikes me that in their gross measurement data, they likely have sufficient numbers with which to compare wild-type (Cre-negative etc.) to conditional het, if they were generating heterozygotes. The authors argue convincingly that loss-of-function drives WSS, but affected individuals are still heterozygotes, so the het genotype is of particular importance.

2) Line 263: "These findings suggest that POGZ is not essential for inhibitory neuron generation."  I think that this is perhaps overstated, since GAD2-Cre would be expected to knock out POGZ in neurons rather than in progenitors. Perhaps the authors could re-word this.

3) I thought that the methodological description of the electrophysiology experiments was too brief. In the methods section, the authors cite another paper by their group, which has details on slice preparation etc. However, it isn't clear how the authors identified excitatory vs. inhibitory neurons in their experiments. Can they add text to describe how neurons were chosen? I think that the channel blockers used in the current paper are not all covered in the other paper either.  Can the authors add info regarding DNQX, D-AP5, strychnine and TTX concentrations etc. to the methods?

4) For Fig. 1D, strictly-speaking, there is no control in the figure. However, GAD-Cre would not delete much in the cortex, whereas Emx1-Cre would not delete in the striatum.  Can the authors either add a control for the reader, or explain better why they don't need a control in the text?

5) Body weight changes are sometimes associated with brain weight/size changes. It would have been nice for brain size to be quantitated in the Emx1 line. Brain size changes seem to differ between POGZ models in the literature.

6) In the statistics section, the authors state that they are running Welch's test throughout the paper, but in Figs. 5 and 6, it says that the stats were done with Student's t test. Can the authors please reconcile?

For the Discussion, I think that the text does not adequately develop a few aspects of the study. 

1) In the discussion, the authors devote relatively little text towards summarizing their own findings. I think it is important to try to capture how the histological data fits with the electrophysiological findings. That is, the authors' data imply that POGZ regulates synaptic activity without affecting neuron number/composition or spine number.

2) It seems to me that the authors spent quite a bit of the discussion talking about POGZ's potential role in mitosis (final paragraph), without discussing how POGZ might act to regulate synaptic transmission.  Would it regulate the expression of synaptic genes? A reader unfamiliar with the POGZ literature might not be able to 'connect the dots'. What do we already know about POGZ and synapses or synaptic gene expression?

2) The authors have not discussed the fact that two papers previously reported similar effects on EPSCs in POGZ mutants. mEPSC frequency was shown to be elevated by Matsumura et al., and IPSC frequency was elevated in Suliman-Lavie et al., albeit in the cerebellum. The authors' findings should be contrasted vs. these results in the discussion.

3) Relatedly, Hamada et al. deleted POGZ either in excitatory or inhibitory neurons, which might be predicted to create an E/I imbalance artificially. However, Matsumura and Suliman-Lavie saw similar features with more global deletions, suggesting that the authors' strategy may have dissected aspects of the POGZ phenotype. I think that this sort of dissection would be the main importance of generating lineage-specific models as the authors have done. I think it is important to address these possibilities in the discussion.

4) Markenscoff-Papadimitriou et a. (PMID: 34879283) should have been cited and discussed. This is another paper about modelling POGZ deficiency in mice.

5) Line 427: "Conversely, the regulatory role of in mitosis suggests its involvement as a transcriptional regulator in the control of neuronal proliferation"  However, neurons do not proliferate.

6) The GAD-POGZ mutant has a phenotype, but it must be admitted that the authors' experiments were not as comprehensive in examining interneurons vs. pyramidal neurons. In particular, the authors never examined morphology or synapse numbers on inhibitory interneurons. They could potentially discuss this out in the text.

Author Response

Reviewer#2

In general, the paper is well written (although the text needs further development in places), and the work is of high quality in general. I have very little to criticize in the authors' manuscript, except that I think it is too short, especially in the Discussion section. I think it should be a good contribution to the field.

> According to the Reviewer’s comment, we totally amended the Discussion section in the in the revised manuscript.

Here are my concerns - all minor:

With respect to the study design, I only spotted a few small issues. 

1) "Control" is not well-defined in the text.  Did the authors pool conditional heterozygotes with wild-type?  If so, are they sure that conditional heterozygotes do not have a phenotype?  It strikes me that in their gross measurement data, they likely have sufficient numbers with which to compare wild-type (Cre-negative etc.) to conditional het, if they were generating heterozygotes. The authors argue convincingly that loss-of-function drives WSS, but affected individuals are still heterozygotes, so the het genotype is of particular importance.

> We used Emx1+/−- and Gad2+/−-Cre mice as control mice in this study. When we previously produced heterozygous (Nestin-cKO+/−) and homozygous (Nestin-cKO−/−) mutations restricted to the nervous system (Reut Suliman-Lavie et al., 2020), we observed a significant reduction in the body weight of Nestin-Pogz cKO−/− mice relative to control littermates across development, with heterozygotes (Pogz cKO+/−) showing an intermediate phenotype (Reut Suliman-Lavie et al., 2020). As to the present model mice, Emx1-Pogz−/− mice, but not Gad2-Pogz−/− mice, showed developmental delay based on the body weight (Fig. 1B). However, we have not further analyzed their heterozygotes. As suggested by the Reviewer, further analyses may clarify if heterozygotes (Emx1-Pogz cKO+/−) show an intermediate phenotype, which are possible to be useful to understand the pathogenicity of POGZ-deficiency.

2) Line 263: "These findings suggest that POGZ is not essential for inhibitory neuron generation."  I think that this is perhaps overstated, since GAD2-Cre would be expected to knock out POGZ in neurons rather than in progenitors. Perhaps the authors could re-word this.

> We agree to the Reviewer’s comment. We deleted the sentence in the revised manuscript, because the observed findings do not necessarily indicate the role of POGZ in neuronal progenitors.

3) I thought that the methodological description of the electrophysiology experiments was too brief. In the methods section, the authors cite another paper by their group, which has details on slice preparation etc. However, it isn't clear how the authors identified excitatory vs. inhibitory neurons in their experiments. Can they add text to describe how neurons were chosen? I think that the channel blockers used in the current paper are not all covered in the other paper either.  Can the authors add info regarding DNQX, D-AP5, strychnine and TTX concentrations etc. to the methods?

> We agree to the Reviewer’s comment. We amended the description and added detailed info on chemicals used in the revised manuscript (p. 4, line 156 - 164). As for the identification of excitatory vs. inhibitory neurons, when we penetrate the cell with a glass capillary, we can empirically know the cell types based on the initial electrophysiological response by cells and the properties, such as membrane capacitance and membrane resistance. We therefore used the word “electrophysiologically” (p.4, line 157 in the revised manuscript). We would greatly appreciate it if the Reviewer could accept this explanation.  

4) For Fig. 1D, strictly-speaking, there is no control in the figure. However, GAD-Cre would not delete much in the cortex, whereas Emx1-Cre would not delete in the striatum.  Can the authors either add a control for the reader, or explain better why they don't need a control in the text?

> We agree to the Reviewer’s comment that there is no strict control in Fig. 1D. According to the reviewer’s suggestion, we provided a more detailed explanation as to why a control may not be necessary (p. 5, line 211 - 214 in the revised manuscript).

5) Body weight changes are sometimes associated with brain weight/size changes. It would have been nice for brain size to be quantitated in the Emx1 line. Brain size changes seem to differ between POGZ models in the literature.

> We agree to the Reviewer’s suggestion that brain weight/size change should be analyzed. Decrease in the body weight of Emx-Pogz mice was statistically significant but also appeared to be relatively small. When we isolated and analyzed 4 Emx-Pogz brains from adult mice (P60), the data were indeed statistically negative. Therefore, we did not further continue the analysis, because we thought it would take many brains to obtain statistically significant data. Considering the time required to collect many adult mouse brains (male and female), we would like to set this up for a future work. We would greatly appreciate it if the Reviewer could accept this explanation.    

6) In the statistics section, the authors state that they are running Welch's test throughout the paper, but in Figs. 5 and 6, it says that the stats were done with Student's t test. Can the authors please reconcile?

> In response to the reviewer's comments, we standardized the statistical analysis method throughout the manuscript using the Mann-Whitney U test and made changes to Figures 1, 5, and 6 in the revised manuscript. Corresponding figure legends have been amended in the revised manuscript.

 7) For the Discussion, I think that the text does not adequately develop a few aspects of the study. 

7-1) In the discussion, the authors devote relatively little text towards summarizing their own findings. I think it is important to try to capture how the histological data fits with the electrophysiological findings. That is, the authors' data imply that POGZ regulates synaptic activity without affecting neuron number/composition or spine number.

We agree to the Reviewer’s suggestion and made some consideration on how the histological data are related to the electrophysiological findings in the Discussion section (p. 14, line 431 - 445).

7-2) It seems to me that the authors spent quite a bit of the discussion talking about POGZ's potential role in mitosis (final paragraph), without discussing how POGZ might act to regulate synaptic transmission.  Would it regulate the expression of synaptic genes? A reader unfamiliar with the POGZ literature might not be able to 'connect the dots'. What do we already know about POGZ and synapses or synaptic gene expression?

> We agree to the Reviewer’s comment. According to the comment, we made descriptions of the roles of Pogz in gene expression.

To determine the mechanism linking chromatin, transcriptional and synaptic dysregulation to cognitive deficits associated with NDD, Pogz-knockdown was performed in prefrontal cortex of juvenile mice (Conrow-Graham et al., 2022). Consequently, the upregulated differentially expressed genes (DEGs) by Pogz- deficiency showed significant under-enrichment of neuronal genes (Conrow-Graham et al., 2022), which was in accordance with previous findings in ASD/ID (Qin et al., 2018; Wang et al., 2021; Voineagu et al., 2011). On the other hand, downregulated DEGs by Pogz-deficiency indicated the decreased transcription of some neuron-related genes. This is also in agreement with the recent discovery that Pogz may promote transcription of clustered synaptic genes in early development (Markenscoff-Papadimitriou, 2021). When the functional roles of the upregulated DEGs was examined using GO pathway analysis to uncover the top biological processes affected by Pogz deficiency, top pathways downregulated by Pogz knockdown were those for action potential and membrane depolarization. We described these issues in the Discussion section in the revised manuscript (p.14, line 464 – p.15, line 487).

7-3) The authors have not discussed the fact that two papers previously reported similar effects on EPSCs in POGZ mutants. mEPSC frequency was shown to be elevated by Matsumura et al., and IPSC frequency was elevated in Suliman-Lavie et al., albeit in the cerebellum. The authors' findings should be contrasted vs. these results in the discussion.

The Reviewer’s suggestion is to the point; comparison of our electrophysiological analyses data with those of the previous works are meaningful. We thus tried to make some discussion. POGZWT/Q1038R mice produced by Matsumura et al. is a model of a de novo Gain-of-Function variant identified in an ASD patient. On the other hand, this study focused on the cerebrum of Emx1- and Gad2-Pogz mice because it is a central cognitive region that is supposed to be impaired in NDDs. Although it is a tough work to compare our present findings to those of the previous work by Matsumura et al., we tried and made some description in the revised manuscript (p. 14, line 446 - 455).   

Meanwhile, Suliman-Lavie et al. performed the analysis using the cerebellar Purkinje cells since POGZ shows significantly increased expression in the cerebellum, and data on cortical neurons are not available. In our present study using Emx1- and Gad2-Pogz mice, POGZ expression should be intact in cerebellum. Althoug it seems a bit difficult to compare the present results to those by Suliman-Lavie et al., we tried to make some discussion (p. 14, line 456 - 463). We would greatly appreciate it if the reviewer could accept the description.

7-4) Relatedly, Hamada et al. deleted POGZ either in excitatory or inhibitory neurons, which might be predicted to create an E/I imbalance artificially. However, Matsumura and Suliman-Lavie saw similar features with more global deletions, suggesting that the authors' strategy may have dissected aspects of the POGZ phenotype. I think that this sort of dissection would be the main importance of generating lineage-specific models as the authors have done. I think it is important to address these possibilities in the discussion.

We agree to the Reviewer’s comment. We described the comparison of our model mice to those by Matsumura and Suliman-Lavie in the Discussion section. We referred to the dissected aspects of the phenotypes as the responses to the Reviewer’s comment as mentioned above. We would greatly appreciate it if the reviewer could accept the description.

7-5) Markenscoff-Papadimitriou et a. (PMID: 34879283) should have been cited and discussed. This is another paper about modelling POGZ deficiency in mice.

We agree to the Reviewer’s suggestion and referred to the work by Markenscoff-Papadimitriou et al. (Ref. 22) in the revised manuscript (p. 15, line 483 - 485).

7-6) Line 427: "Conversely, the regulatory role of in mitosis suggests its involvement as a transcriptional regulator in the control of neuronal proliferation"  However, neurons do not proliferate.

We agree to the Reviewer’s comment and changed the words “neuronal proliferation” to “progenitor proliferation” in the revised manuscript (p. 13, line 424).

7-7) The GAD-POGZ mutant has a phenotype, but it must be admitted that the authors' experiments were not as comprehensive in examining interneurons vs. pyramidal neurons. In particular, the authors never examined morphology or synapse numbers on inhibitory interneurons. They could potentially discuss this out in the text.

We agree to the Reviewer’s suggestion and added a description in the revised manuscript (p. 14, line 441 - 445).

Round 2

Reviewer 1 Report

Comments and Suggestions for Authors

Authors corrected the issues raised in regard the statistical analysis all along the manuscript. I believe that the manuscript can be published in its current form.